# p38 MAPK in Glucose Metabolism of Skeletal Muscle: Beneficial or Harmful?

**DOI:** 10.3390/ijms21186480

**Published:** 2020-09-04

**Authors:** Eyal Bengal, Sharon Aviram, Tony Hayek

**Affiliations:** 1Department of Biochemistry, Rappaport Faculty of Medicine, Technion-Israel Institute of Technology, P.O. Box 9649, Haifa 31096, Israel; avirams@technion.ac.il; 2Department of Internal Medicine E, Rambam Medical Center, Haifa 31096, Israel; t_hayek@rambam.health.gov.il

**Keywords:** skeletal muscle, energy metabolism, signal transduction, p38 MAPK, exercise, type 2 diabetes

## Abstract

Skeletal muscles respond to environmental and physiological changes by varying their size, fiber type, and metabolic properties. P38 mitogen-activated protein kinase (MAPK) is one of several signaling pathways that drive the metabolic adaptation of skeletal muscle to exercise. p38 MAPK also participates in the development of pathological traits resulting from excessive caloric intake and obesity that cause metabolic syndrome and type 2 diabetes (T2D). Whereas p38 MAPK increases insulin-independent glucose uptake and oxidative metabolism in muscles during exercise, it contrastingly mediates insulin resistance and glucose intolerance during metabolic syndrome development. This article provides an overview of the apparent contradicting roles of p38 MAPK in the adaptation of skeletal muscles to exercise and to pathological conditions leading to glucose intolerance and T2D. Here, we focus on the involvement of p38 MAPK in glucose metabolism of skeletal muscle, and discuss the possibility of targeting this pathway to prevent the development of T2D.

## 1. Introduction

### 1.1. Skeletal Muscle Energy Metabolism

Skeletal muscle comprises about 40% of the total body mass and accounts for around 30% of resting metabolic rate [1]. Due to its role in locomotion, this tissue is a major energy consumer, especially during exercise. Thus, it is the primary site of glucose disposal and the major glycogen storage organ and is, therefore, absolutely critical for glycemic control and the metabolic homeostasis of the body [2]. Since ATP’s resting intramuscular stores are small, energy needs to tightly regulate the metabolic pathways that generate ATP to maintain ATP at constant levels. ATP is generated in the muscle by the oxidation of carbohydrates and lipids. The primary carbohydrate energy source in skeletal muscle is glucose that is processed from internal glycogen stores (glycogenolysis) or is extracted from the blood.

#### 1.1.1. Glucose Uptake

Glucose is transported by the Glut family of membrane transporters, with Glut1 and Glut4 being the major transporters in skeletal muscle. Whereas Glut1 is responsible for the basal uptake of glucose, Glut4 is an inducible transporter that facilitates glucose uptake by insulin or muscle contraction. In the basal state, Glut4 is mainly associated with intracellular vesicles lying adjacent to the plasma membrane of the muscle fiber (sarcolemma). Insulin and muscle contraction facilitate the translocation of vesicular Glut4 to the sarcolemma and the T tubular system where Glut4 transports glucose from the blood into the cytoplasm of muscle cells. Distinct molecular mechanisms are responsible for Glut4 translocation into the sarcolemma by contraction and insulin [3]. However, the two pathways converge at the GTPase activating proteins (GAPs), TBC1D1, and TBC1D4 (AS160). Phosphorylation of TBC1D1/TBC1D4 at particular residues by either AMP-activated protein kinase (AMPK) in contracting muscle or by insulin-activated Akt inhibits their GTPase-activating domain and enables the exchange of GDP to GTP–bound state of Rab. Once activated, Rab mediates the translocation of vesicular Glut4 to the plasma membrane and facilitates glucose uptake [3]. Hereafter, we will describe the role of p38 mitogen-activated protein kinase (MAPK) in glucose uptake under different physiological conditions.

#### 1.1.2. Fiber Type and Glucose Metabolism

Skeletal muscle fibers are heterogeneous with respect to their contractile apparatus and metabolism [4]. Three types of motor units, called slow twitch, fast fatigable, and fast fatigue-resistant, are composed of type 1, 2A, 2B, and 2X fibers. These fibers are distinguished by their myosin heavy chain (MyHC) isoforms and their oxidative/glycolytic metabolism. Whereas type 1 and 2A fibers are rich in mitochondria and derive their energy mostly from oxidative metabolism, the mitochondria-poor type 2B and 2X fibers primarily utilize anaerobic glycolysis for ATP production. Different muscles contain particular ratios of fast and slow-twitch fibers. The skeletal muscle is a highly plastic organ that adapts to the changes in metabolic needs. Whereas physical inactivity and obesity induce a phenotypic shift from oxidative type 1 and 2A fibers to glycolytic type 2B fibers [5], endurance exercise causes a switch from glycolytic type 2B fibers towards more oxidative type 1 and 2A fibers [6].

### 1.2. P38 MAPK Is Involved in Various Aspects of Whole-Body Energy Metabolism

Four isoforms of p38 MAPK, alpha, beta, gamma, and delta, are encoded by four different genes and have different tissue expression patterns [7]. These kinases are members of a larger family that includes at least three additional kinases; the extracellular signal-regulated kinases (ERK1/2), ERK5 (also known as BMK1), and the Jun amino-terminal kinases (JNK1-3). P38 and JNK are considered stress-activated protein kinases (SAPKs) that participate in the cellular response to metabolic and other stress conditions. In these roles, they are likely to contribute to metabolism-related pathogenesis. The four isoforms of p38 are activated by environmental and genotoxic stress conditions, inflammatory cytokines, and hormones that drive dual phosphorylation in the activation loop sequence Thr-Gly-Tyr. The primary activating kinases (MAPKK) of the four-p38 isoforms are MKK3 and MKK6. In turn, p38 isoforms phosphorylate and activate many substrates, including proteins and enzymes of glucose and lipid metabolism. Below is a list of p38 MAPK targets, mostly transcription factors involved in the energy metabolism of different tissues.

#### 1.2.1. PPAR Gamma Activator-1 Alpha (PGC1α)

PGC1α is a master metabolic co-factor of mitochondrial biogenesis. It is involved in respiration and oxidative phosphorylation, Glut4 expression, conversion of type 2B muscle fibers to type 2A and 1 fibers and in gluconeogenesis in the liver. PGC1α is a transcriptional coactivator whose phosphorylation by p38 prevents the binding of the repressor p160MBP and enables its activity in adipose tissue, muscle, and liver [8]. p38 MAPK also phosphorylates and activates the ATF2 and Mef2 transcription factors, which in turn upregulate the expression of PGC1α. Hence, p38 MAPK positively affects both the transcription and activity of PGC1α.

#### 1.2.2. Activating Transcription Factor 2 (ATF2)

ATF2 is a transcription factor, a member of the leucine zipper (bZip) family, binds to a cAMP-responsive element (CRE) as homodimer or heterodimer with c-Jun. ATF2 is usually phosphorylated and activated by SAPKs, p38, and JNK [9]. P38 MAPK regulates brown adipose tissue differentiation by increasing the expression levels of uncoupling protein 1 (UCP1) through the phosphorylation of ATF2 and PGC1α [10,11,12]. Inhibition of p38 MAPK in high-fat diet (HFD)-fed mice prevented obesity and alleviated insulin resistance [11].

#### 1.2.3. CCAAT/Enhancer-Binding Protein α (C/EBPα)

This bZip transcription factor binds to DNA as a homodimer, or as a heterodimer with the related proteins C/EBPβ and C/EBPγ, as well as with distinct transcription factors such as c-Jun. It is a crucial regulator of adipogenesis, accumulation of lipids in those cells, and the metabolism of glucose and lipids in the liver [13]. P38 MAPK phosphorylates Ser21 of C/EBPα, which regulates whole-body glucose homeostasis [14]. Serine phosphorylation by p38 enhances C/EBPα activity necessary for the transcription of phosphoenolpyruvate carboxykinase (PEPCK), a rate-limiting enzyme in liver gluconeogenesis [15]. P38 MAPK is also involved in adipocyte differentiation through the phosphorylation of C/EBPβ, which induces peroxisome proliferator-activated receptor gamma (PPARγ).

#### 1.2.4. cAMP Response Element-Binding Protein (CREB)

Phosphorylation of CREB at Ser133 by several kinases induces its transcriptional activity [16]. P38 MAPK does not directly phosphorylate CREB, but rather induces CREB phosphorylation via MSK1 (mitogen-and stress-activated kinase 1). P38 MAPK augments liver gluconeogenesis via the CREB protein that activates the transcription of *Pepck*, *Glucose 6-phosphatase*, and *Pgc1α* genes [10,17,18].

#### 1.2.5. Glycogen Synthase (GS)

It is the rate-limiting enzyme in the glycogen biosynthesis pathway. The β isoform of p38 has been shown to phosphorylate GS and inhibit its activity and the conversion of glucose to glycogen in the liver [19,20].

#### 1.2.6. Peroxisome Proliferator-Activated Receptor α (PPARα)

PPARα belongs to a group of nuclear receptor proteins that function as transcription factors as heterodimers with the retinoid X receptor (RXR). P38 MAPK promotes the β oxidation of fatty acids in the liver and cardiomyocytes through the phosphorylation of PPARα [21,22]. p38 MAPK also inhibits hepatic lipogenesis [23], and therefore, when activated, it prevents lipid accumulation in the liver.

#### 1.2.7. X-Box Binding Protein 1 (Xbp1)

This member of the CREB/ATF family of transcription factors which is activated by splicing under conditions of endoplasmic reticulum (ER) stress, is a master regulator of unfolded protein response (UPR) folding capacity. Obesity and T2D increase ER stress in the liver, and expression of Xbp1 significantly decreases ER stress, restores glucose tolerance, and reduces blood glucose levels [24]. By phosphorylating Thr48 and Ser61 residues of the spliced form of Xbp1, p38 MAPK enhances its nuclear translocation and activity. P38 MAPK activity is diminished in the livers of obese mice compared with lean mice. Conversely, its activation by the expression of constitutively active MKK6 kinase (MKK6(E)) reduces ER stress and establishes euglycemia in obese diabetic mice [25].

## 2. Regulation of Glucose Metabolism by p38 MAPK in the Adaptation of Skeletal Muscle to Exercise

It is well established that physical activity and exercise training are beneficial to health and can prevent insulin resistance, type 2 diabetes (T2D) [26], and sarcopenia [27]. Muscle adaptation to exercise includes changes in contractile proteins, mitochondrial function, metabolic regulation, and specific signaling pathways that regulate gene expression. Muscle adaptation is necessary, since maximal exercise induces a 20-fold increase in whole-body metabolism and up to a 100-fold increase in ATP consumption relative to resting skeletal muscle [28]. Exercise affects aerobic (endurance) and anaerobic (resistance) metabolism, with different benefits to each modality, as is reviewed in detail elsewhere [28]. The combination of both types of exercise is more effective than each in preventing obesity, reducing insulin resistance, and improving glycemic control in T2D [29]. Here, we focus on p38 MAPK in the adaptation of skeletal muscle carbohydrate metabolism to exercise. Three of the four isoforms of p38, α, β, and γ are expressed in skeletal muscle [30]. Exercise induces their activation, whose level and duration depend on the mode of exercise [31,32,33]. P38 MAPK affects the crucial processes necessary for the adaptation to the metabolic demands and energy needs of the exercising skeletal muscle. This is done by increasing glucose transport into the tissue, elevating glycolytic and citric acid cycle flux, and raising the mitochondria’s number and functional quality. P38 MAPK mediates these positive effects by phosphorylating diverse transcription factors and coactivators involved in carbohydrate metabolism.

### 2.1. Glucose Uptake

Exercise induces a dramatic increase in glucose uptake by the Glut4 transporter. The primary sensor of energy demand in contracting muscle is AMP-activated protein kinase (AMPK), a serine/threonine kinase activated by low energy reservoirs and muscle contraction. Low intracellular energy levels are reflected by increased AMP: ATP ratio and muscle contraction by increasing the concentration of cytoplasmic Ca^2+^. The alpha subunit of the heterotrimeric AMPK is phosphorylated at Thr172 by liver kinase B1 (LKB1), by calcium/calmodulin-dependent protein kinase β (CaMKβ), and by the transforming growth factor-beta activated kinase 1 (TAK1) that dramatically increases its kinase activity towards metabolic substrates [34]. Among these substrates are the Rab GTPase-activating proteins (GAPs), TBC1D1 and TBC1D4 (AS160) that are inactivated by phosphorylation, a step necessary for Glut4 translocation to the plasma membrane. A possible role of p38 MAPK in glucose uptake was deduced from the findings that its activity is positively correlated with and depends on AMPK [35,36,37,38]. Activation of both kinases, AMPK and p38 MAPK by TAK1 may explain their synchronization. Moreover, the Mef2 transcription factors involved in the expression of Glut4 in the exercised muscle are substrates of p38 MAPK [39,40]. Endurance exercise increases *Glut4* gene expression by the binding of Mef2 to its cognate site at the *Glut4* promoter [41]. Exercise also increases the kinase activity of p38α/β, which associates with and phosphorylates Mef2 to enhance its transcriptional activity. Indeed, Chambers and colleagues showed that stretch-stimulated glucose uptake was diminished by inhibitors of p38α and β [35,42]. However, other studies investigating adipocytes and myotubes revealed an unexpected off-target activity of the p38α, β inhibitor SB203580, that directly bound to Glut4 transporter and interfered with its activity. The authors concluded that this inhibitor could diminish glucose uptake through the inhibition of Glut4 and not necessarily through p38 MAPK [43,44]. Nevertheless, the addition of other p38α, β inhibitors that are chemically unrelated to SB203580, and the expression of a dominant-negative p38α mutant (p38AGF) reduced insulin-stimulated glucose uptake [45]. In another study, activation of p38 MAPK by expressing constitutively active MKK6 (MKK6E) upregulated Glut1 and down-regulated Glut4, thereby increased basal glucose uptake, but diminished insulin-mediated glucose uptake in L6 myotubes [46]. Studies investigating intracellular and extracellular agonists of p38 MAPK implicate this kinase in insulin-independent glucose uptake by muscles. For example, the drug anisomycin, which activates the stress kinases JNK and p38 MAPKs, increased glucose uptake in resting muscles [47]. Homocysteine sulfonic acid, a glutamate receptor agonist, was shown to stimulate glucose uptake in myotubes through an AMPK-p38 -dependent pathway [48]. Rac1 GTPase, one of the known effectors of p38, facilitates exercise mediated translocation of Glut4 to the plasma membrane and glucose uptake [49]. Furthermore, the accumulation of cytosolic reactive oxygen species (ROS) produced by NADPH oxidase 2 (NOX2), entailed phosphorylation of p38 MAPK, and stimulated Glut4-dependent glucose uptake in muscle during exercise [50]. However, p38 MAPK phosphorylation levels were not always consistent with the reduction in glucose uptake in muscles of mice devoid of NOX2 activity, indicative that p38 may not be the mediator of glucose uptake, due to the accumulation of cytoplasmic ROS. Investigation of the role skeletal muscle’s most abundant isoform, p38γ, in glucose transport revealed that its overexpression in mature muscle fibers reduced the expression of Glut4 and decreased contraction-induced glucose uptake [51]. Overall, most studies demonstrated the involvement of p38 MAPK activity in exercise-mediated glucose uptake. Yet, it remains unclear whether this family of kinases mediates glucose uptake by inducing the expression of Glut4, Glut1, or both, and the particular roles of the different isoforms in basal and stimulated glucose uptake.

### 2.2. Mitochondrial Activities

The transcriptional coactivator PGC1α is a crucial regulator of mitochondrial function, oxidative metabolism, and energy homeostasis in a variety of tissues [52]. The expression of PGC1α is induced in muscle following endurance or resistance exercise [53]. Forced expression of PGC1α in mice’s skeletal muscle is sufficient to improve performance capacity during exercise [54] and protect skeletal muscle from sarcopenia and aging-related metabolic diseases [55]. The core function of PGC1α in skeletal muscle is to increase the number and capacity of mitochondria and enhance oxidative metabolism. It orchestrates these functions by co-activing the estrogen-related receptor α (ERRα), the nuclear respiratory factor 1 and 2 (NRF1 and 2), and subsequently increasing the level of the mitochondrial transcription factor A (TFAM). P38 MAPK plays a central role in the expression and activity of PGC1α. Several inducible transcription factors rapidly and robustly increase PGC1α gene expression; CREB and ATF2 [56], that bind to cAMP-response elements (CRE), and Mef2C and D [57], that bind to the YTA(A/T)_4_TAR sequence found within regulatory sequences of the *Pgc-1α* gene. These transcription factors are phosphorylated and activated by p38 MAPK and by Ca^2+^ signaling mediated by calcineurin and CaMK pathways [58,59]. Therefore, several signaling pathways, including p38 MAPK, converge onto *Pgc1α* gene expression and muscle adaptation following exercise. Moreover, transgenic mice with skeletal muscle-specific expression of active MKK6 express higher levels of PGC1α and other markers of mitochondrial biogenesis in fast-twitch muscle [56]. At another level, p38 MAPK directly phosphorylates the PGC-1α protein to promote its stability and activity [8]. PGC1α contains a negative regulatory domain that inhibits the function of the transcriptional activation domain [60]. Transcription suppression is relieved when p38 MAPK phosphorylates residues Thr262, Ser265, and Thr298 within the negative regulatory domain of PGC1α [61,62]. These phosphorylation events stabilize PGC1α and drive its dissociation from the p160MBP repressor [8]. Forced expression of PGC1α in skeletal muscle adapting to exercise also leads to the conversion of muscle fibers from type 2 (fast-twitch) to type 1 (slow-twitch) [63]. Moreover, an autoregulatory loop established between Mef2 and PGC1α maintains type 1 fibers in the contracting muscle [64]. Although there is no direct supporting evidence, it is reasonable to conjecture that p38 MAPK is also involved in fiber type adaptation by activating Mef2 and PGC1α. However, some results suggest otherwise; muscle-specific deletions of each p38 MAPK isoform (α, β, γ) indicated that none of the isoforms were required for exercise-induced slow fiber type transformation [65]. Still, p38γ MAPK was the only isoform required for endurance exercise-induced mitochondrial biogenesis. Overexpression of a dominant-negative form of p38γ MAPK, but not of p38α MAPK or p38β MAPK, blocked contraction-induced *Pcg-1α* transcription [65]. It was, therefore, concluded that p38γ MAPK was involved in metabolic adaptation, but not in the adaptation of skeletal muscle contractile machinery to exercise. Interestingly, the results of another study indicate that p38γ MAPK is phosphorylated and activated in slow (soleus) but not in fast (gastrocnemius) muscles. Loss of p38γ MAPK reduced slow myosin-expressing fibers and increased the number of fast myosin-expressing fibers in the soleus [66]. Overall, these studies suggest that the p38 MAPK pathway is involved in the adaptation of mitochondria and oxidative metabolism to exercise. Still, its involvement in the adaptation of the contractile machinery calls for further inquiry. Future studies should also look for possible contributions of different p38 MAPK isoforms to the metabolic adaptation of muscle to exercise.

## 3. P38 MAPK in the Development of Insulin Resistance and Type 2 Diabetes (T2D)

### 3.1. Insulin Resistance

Insulin resistance is defined as the reduction in insulin’s ability to stimulate glucose uptake by the body’s peripheral tissues. Insulin is known to activate the canonical IRS-PI3K-Akt pathway and phosphorylate and inactivate Akt substrate of 160 kDa (AS160, TBC1D4). By inactivating AS160, insulin promotes Glut4-intracellular vesicles’ transport and their fusion with the plasma membrane, consequently increasing glucose uptake by tissues [3]. Obesity and genetic factors induce chronic metabolic syndrome that involves dyslipidemia, inflammation, hypertension, and insulin resistance. Systemic insulin resistance triggers chronic hyperglycemia, which causes pancreatic β cells to secrete more insulin. At later stages of the syndrome, increased insulin secretion induces ER stress, β cell death, insulin deficiency, and diabetes. Skeletal muscle accounts for ~30% of the resting metabolic rate in humans [1,67], and up to 80% of glucose disposal under insulin-stimulated conditions, and it is thus the primary organ regulating glucose balance [1,2]. One model explaining the development of insulin resistance in skeletal muscle argues that the excess of lipids stored in the adipose tissue are released into the circulation as fatty acids that are taken up and accumulate in organs, such as the liver and the skeletal muscles [68]. Intramuscular lipid metabolites damage mitochondrial activity and consequently increase the production of reactive oxygen radicals. Another model predicts that calorie surplus imposes maximal mitochondrial respiration and leakage of electrons from the electron transport chain, forming oxygen radicals [69]. Oxidative stress inhibits insulin action via activation of serine/threonine kinases, including p38 MAPK that phosphorylate and neutralize insulin receptor substrate 1 (IRS1) [70,71]. Excessive oxygen radicals also inflict cellular damage by oxidizing proteins, fatty acids, and carbohydrates.

### 3.2. P38 MAPK in Skeletal Muscle Insulin Resistance

The role of the p38 MAPK pathway in the development of insulin resistance remains controversial. Likely sources of the dispute are different experimental-settings and systems, some of which investigated insulin resistance in muscle cell cultures and others in animal models. The general concept for the involvement of p38 MAPK in insulin resistance is that its activity inhibits insulin signaling by enhancing inhibitory phosphorylation of IRS1 at Ser307 and other related residues [72]. IRS1 may not be a direct target of p38 MAPK, but of other associated kinases, including IkappaB kinase (IKK), JNK, and some novel atypical PKC isoforms. The treatment of primary myotubes with tumor necrosis factor (TNF)α produced insulin resistance associated with Ser307 phosphorylation, which was mediated by p38 α,β MAPK [73]. Analysis of oxidative stress-induced insulin resistance in soleus muscle strips revealed an exciting insight into the possible role of p38 MAPK [74]. Oxidative stress-induced chronic activity of p38 MAPK increased basal glucose transport activity (in the absence of insulin), but blocked, at the same time, insulin signaling and, consequently, insulin-induced glucose transport [75]. In another study, forced expression of constitutive active MKK6/3 in L6 myotubes increased the expression of Glut1 while decreasing that of Glut4, thereby enhancing basal glucose transport and diminishing insulin-induced transport [46]. In this respect, it is interesting to note that the basal phosphorylation of p38 MAPK was higher in the skeletal muscle of type 2 diabetic human subjects than in healthy controls. Insulin treatment transiently increased phosphorylation of p38 MAPK in skeletal muscles of non-diabetic individuals, but not of T2D patients [76]. Ex vivo treatment of isolated soleus muscle with oxidative radicals entailed a significant decrease in insulin-stimulated glucose transport activity associated with selective loss of IRS1 and IRS2 proteins and augmented phosphorylation of IRS1 at Ser307. Inhibition of p38α/β MAPK partially restored the impaired insulin-stimulated glucose transport activity [77]. Together, these findings suggest a role of p38 MAPK in the mediation of basal glucose transport and in the negative regulation of insulin-stimulated glucose transport activities that may be regulated by different isoforms of p38 (α, β or γ) MAPK. At least one study pointed at p38γ MAPK as the isoform mediating both activities, inducing basal glucose uptake, and inhibiting muscle contraction-stimulated glucose uptake [51]. An additional aspect of the involvement p38 MAPK in insulin resistance is its suppression of PGC1α *α* and PGC1β expression that entails the accumulation of lipid metabolites [78]. This result is unexpected given the reported role of p38 MAPK in increasing PGC1α expression/function in contracting muscles. Insulin resistance associated with the elevated phosphorylation of JNK and p38 MAPKs was also observed in rat skeletal muscles after 6 h of hind limb immobilization [79]. Inflammatory cytokines such as TNFα and Il-6 also cause insulin resistance, and p38 MAPK itself is a crucial mediator of the expression of genes encoding for proinflammatory cytokines in skeletal muscles. Indeed, inhibition of p38 MAPK in myotubes derived from T2D subjects prevented proinflammatory cytokines’ secretion, but did not improve insulin resistance [80]. In another study, the treatment of primary myotubes with TNFα produced insulin resistance dependent on the elevated activity of p38 MAPK [73]. TNFα-P38 MAPK pathway induced serine phosphorylation of insulin receptor (IR) and IRS1, and reduced tyrosine phosphorylation of the same molecules. P38 MAPK-dependent insulin resistance was also reported in myotubes that were cultured in conditioned medium of macrophages treated with the saturated fatty acid, palmitate, and contained increased levels of proinflammatory cytokines [81]. In conclusion, different metabolites known to be involved in the development of insulin resistance (saturated fatty acids, oxidative radicals, and inflammatory cytokines), induce the chronically elevated p38 MAPK in skeletal muscles, that is also likely to mediate insulin resistance.

Despite the above studies that establish a role for p38 MAPK in insulin resistance, its requirement for oxidative metabolism and its possible function in the transition of fast to slow-twitch fibers during exercise is in line with the idea that elevated p38 MAPK activity might prevent the development of obesity and insulin resistance. Indeed, an accepted perception is that insulin-stimulated glucose transport is higher in muscles enriched with slow-twitched oxidative fibers [82]. Several animal and cell culture studies support a role for p38 MAPK in preventing insulin resistance development. By manipulating the expression of MAPK phosphatase-1 (MKP-1) that inactivates both JNK and p38 MAPK in mouse skeletal muscle, Bennett and colleagues suggested that the activities of p38 MAPK and JNK were required for whole-body energy expenditure [22,83,84]. High-fat diet (HFD) nutrition of mice that upregulated the expression of MKP-1 in skeletal muscle entailed obesity and insulin resistance through the inactivation of p38/JNK MAPKs. Conversely, knockout of MKP-1 in skeletal muscles of mice fed on HFD prevented the development of insulin resistance [83]. These authors suggested that increased p38/JNK activity in skeletal muscle prevents obesity and insulin resistance by augmenting oxidative metabolism. However, all three MAPKs, ERK, JNK, and p38 targeted by MKP-1, can affect metabolic regulation. For example, mice lacking ERK1 display resistance to diet-induced obesity, coupled with protection from insulin resistance [85]. Thus, muscle expression of MKP-1 probably reflects complex changes in the activities of all three major classes of MAPK. Adiponectin, the most abundant peptide secreted from adipocytes, is also secreted by additional tissues, including skeletal muscle. Adiponectin has many metabolic benefits and improves glucose uptake, utilization, and fatty acid oxidation in myotubes [86]. The peptide binds to its receptors AdipoR1, and AdipoR2 found on membranes of skeletal muscle cells, and induces AMPK and p38 MAPK [87]. Activation of p38 MAPK and AMPK is essential for adiponectin–induced glucose uptake and fatty acid oxidation. These two kinases mediate metabolic changes by stimulating the transcriptional activity of PPARα [88]. Moreover, the forced expression of APPL1, an intracellular adaptor protein of adiponectin signaling, enhances phosphorylation of AMPK and p38 MAPK in myotubes, leading to the translocation of Glut4 to the plasma membrane [89]. By activating the two kinases, AMPK and p38 MAPK, a conserved 13 residue-long peptide derived from the adiponectin collagen domain (ADP-1) was sufficient to induce glucose uptake in TNFα-treated insulin-resistant myotubes [90].

Although the common idea that p38 MAPK is involved in insulin-independent glucose uptake, several studies also suggest its involvement in insulin-dependent glucose uptake. First, p38 MAPK was transiently phosphorylated, not only by muscle contraction, but also by the administration of insulin [91,92,93]. Moreover, exercise increased p38 MAPK phosphorylation, and insulin administration following exercise induced even higher p38 phosphorylation levels that were correlated with enhanced insulin sensitivity in human skeletal muscles [93]. A more recent study that tested the hypothesis that exercise increases insulin sensitivity revealed that anisomycin-mediated activation of p38 in skeletal muscles increased insulin-mediated glucose transport. However, the same study also showed that p38 was not necessary to increase insulin sensitivity following muscle contraction [47]. In this respect, it is essential to mention the early report by Klip and colleagues that SB203580, an inhibitor of p38 α and β attenuated insulin-stimulated glucose uptake [42]. However, later studies by the same group ruled out the involvement of p38 in the process [43]. Moreover, SB203580 reduced insulin-mediated glucose transport through the inhibition of Glut4 transporter activity in a mechanism that might involve direct interaction of the inhibitor with Glut4 that maintained the transporter in an inactive conformation, or through the removal of an inhibitory protein [44,94,95].

In summary, under conditions that affect insulin resistance, chronically activated p38 MAPK is involved with additional kinases, in the inactivation of IRS molecules. Interestingly, despite its role in preventing insulin-mediated glucose uptake, p38 MAPK may increase basal and insulin-independent glucose uptake by skeletal muscle. Different treatments/conditions that induced a burst of p38 MAPK activity prevented obesity and insulin resistance. These conditions include muscle contraction and hormones like adiponectin. Therefore, whereas p38 MAPK prevents insulin-mediated glucose transport, it participates with other signaling molecules in insulin-independent glucose uptake. Acute activation of p38 MAPK may also play a role in insulin-dependent glucose uptake. The intensities and duration of p38 MAPK signals, the involvement of specific p38 MAPK isoforms, and the molecular mechanisms of modulation of glucose uptake await future investigation.

### 3.3. P38 MAPK as a Potential Target for Prevention of Obesity-Induced T2DM

The double edged-sword activity of p38 MAPK in the metabolism of skeletal muscle raises the question of what intervention with this kinase or pathway could beneficially prevent the development of T2DM. On the one hand, elevation of p38 MAPK activity is expected to increase skeletal muscle oxidative metabolism and glucose uptake. To date, regular exercise is more effective than pharmacological intervention in the treatment and prevention of T2DM [26] and sarcopenia [27]. P38 MAPK mediates stimulation of glucose uptake during exercise by elevating the expression levels of Glut4 and increasing PGC1α activity that entails mitochondrial biogenesis and oxidative metabolism. These positive effects of p38 MAPK are insulin-independent, and therefore can bypass insulin resistance. For decades, the most important first-line drug for T2D patients’ treatment is metformin, which induces the activation of AMPK [96]. AMPK facilitates the translocation of Glut4 to the plasma membrane and increases PGC1α-mediated mitochondrial biogenesis and respiration. The concurrent activation of AMPK and p38 MAPK signaling during exercise indicates that the two pathways cross-react and synergize in increasing glucose uptake and oxidative metabolism in skeletal muscle. Interestingly, forced activation of p38 MAPK in livers of obese mice reduces endoplasmic reticulum (ER) stress and establishes euglycemia [25]. Therefore, the application of pharmacoactive agents that augment p38 MAPK activity together with AMPK agonists may synergistically improve glucose import and oxidative metabolism of skeletal muscle in an insulin-independent fashion. However, activation of p38 MAPK to treat obesity and T2D holds a risk due to its possible deleterious effects on whole-body metabolism. For example, the elevated chronic activity of p38α MAPK and p38β MAPK that result from inactivity, denervation, aging (sarcopenia), or cancer (cachexia) participates in catabolism and muscle atrophy and weakness. Under these conditions, p38 MAPK upregulates E3 ligases like MAFbx/Atrogin1 and Murf1 targeting proteins for proteasome degradation [97,98,99,100,101]. Inhibition of p38 MAPK activity protects muscle from the oxidative damage and prevents proteolysis, and was, therefore, suggested as a potential target for the treatment of muscle atrophy [102,103,104]. The continuous activity of p38 MAPK is also involved in the etiology of inflammatory diseases and specifically bowel diseases, like ulcerative colitis and Crohn’s disease [105]. P38 induces the expression of proinflammatory cytokines like TNFα and Il1-β, and the inhibition of p38 MAPK can effectively suppress the expression of circulating inflammatory mediators.

An alternative option for preventing insulin resistance is the inhibition of p38 MAPK activity. This approach’s primary rationale is prevention of the inhibitory phosphorylation of IRS molecules that p38 MAPK mediates and ensuing the restoration of insulin signaling. Moreover, inhibition of p38 MAPK should block hepatic gluconeogenesis and to consequently reduce hepatic glucose release and lower blood glucose levels [10,23]. Interestingly, the deletion of the *mapk14* gene encoding for p38α MAPK in the liver is associated with elevated levels of active AMPK, which is known to suppress gluconeogenesis [106]. In fact, metformin’s significant therapeutic effect is the suppression of liver gluconeogenesis, likely by the activation of AMPK [96]. Therefore, some of the beneficial effects of p38 MAPK inhibition in reducing blood glucose levels may be mediated by liver AMPK. However, as mentioned before, since p38 MAPK activity is vital for mitochondrial function and oxidative metabolism, its inhibition may have adverse effects on muscle glucose metabolism [8,56,62]. Inhibition of p38 MAPK is anticipated to reduce skeletal muscle oxidative phosphorylation and compensate for the lost energy by increasing anaerobic glycolysis [102]. The prevailing model is that excessive calorie intake cause mitochondrial electron leak, accumulation of ROS, mitochondrial dysfunction, and insulin resistance [68,107,108]. Studies in recent years showed that insulin resistance occurred before the deterioration of mitochondrial function [109,110]. Therefore, the loss of mitochondrial oxidative metabolism may not be the cause of insulin resistance. One approach for preventing insulin resistance and improving glucose/fat metabolism is to increase slow/oxidative muscles [111,112]. An equally effective different approach is to increase the relative proportion of glycolytic muscle fibers, which were proven to reduce fat mass and increase insulin sensitivity [113,114,115,116]. By increasing the glucose uptake rate from the plasma and their glycolytic flux, fast-glycolytic muscles maintain energy levels similar to those of slow-oxidative muscles. In fact, the commonly prescribed drugs to treat T2D, biguanides such as metformin, inhibit complex I of the respiratory chain and thereby impair both mitochondrial function and cell respiration [117,118]. Therefore, increasing glycolytic metabolism in skeletal muscle by inhibiting p38 MAPK may help restore glucose tolerance in diabetic patients.

## 4. Conclusions

Despite the substantial progress in understanding p38 MAPK involvement in the regulation of glucose metabolism in skeletal muscle, there remain open questions concerning the identities and involvement of p38 MAPK isoforms in different physiologic states and their modes of affecting their metabolic targets. Due to its massive energy consumption, skeletal muscle functionality is critical for maintaining whole-body glucose homeostasis. Insulin resistance and sarcopenia develop mainly in obese and old immobile individuals due to dysregulated energy metabolism of skeletal muscles, and p38 MAPK is involved in the development of these maladies. Hence, depending on the physiological context, p38 MAPK activity may lead to harmful consequences of sarcopenia and insulin resistance in the immobile muscle or beneficial effects of increased glucose sensitivity and metabolism in the contracting muscle. This review summarized the involvement of p38 MAPK in glucose metabolism of skeletal muscle under the two extreme conditions of exercise (health) and obesity (disease), as schemed in Figure 1. In exercise training, muscle contraction utilizes an enormous amount of energy, supplied by the adaptation of muscles to glycolytic (resistance) or oxidative (endurance) glucose metabolism. Under these conditions, p38 MAPK facilitates glucose transport in an insulin-independent manner and improves insulin-dependent glucose transport by inducing transcription of genes encoding the glucose transporters, Glut1, and Glut4. P38 MAPK also increases oxidative metabolism by inducing the transcription and the activity of PGC1α, the “master coactivator” of mitochondrial biogenesis and activity. Interestingly, in the contracting muscle, p38 MAPK is co-expressed with AMPK, the “energy sensor” of the cell. The activities of these two kinases synergistically increase glucose import in a pathway alternative to that of insulin. They also facilitate together mitochondrial oxidative metabolism to provide the necessary amount of energy for muscle contraction.

In obesity, however, the activity of p38 MAPK contributes to the development of insulin resistance. The surplus in skeletal muscle calorie intake and the elevated levels of circulating fatty acids that penetrate muscle cells, increase intramuscular fat metabolites that reduce mitochondrial oxidative capacity and high electron leakage that generate ROS. ROS and lipid metabolites activate some kinases, including p38 MAPK, that impair the insulin-signaling pathway by phosphorylating serine/threonine residues of IRS1 and preventing tyrosine phosphorylation and activation of IRS1 by insulin. Consequently, insulin signaling is impaired and Glut4-mediated glucose uptake is prevented. The chronic activity of p38 MAPK also decreases the expression of PGC1 proteins and, consequently, diminishes glucose utilization by the mitochondria. Under these pathological conditions, p38 MAPK activates the *Glut1* gene expression and increases the basal diffusion of glucose independently of insulin signaling.

In sum, the p38 MAPK pathway is a double-edged sword that increases insulin-independent glucose uptake and mitochondrial oxidative phosphorylation in a healthy lifestyle while inhibiting, in unhealthy lifestyles, the same processes mediated by insulin signaling, leading to metabolic syndrome.

## Figures and Tables

**Figure 1 ijms-21-06480-f001:**
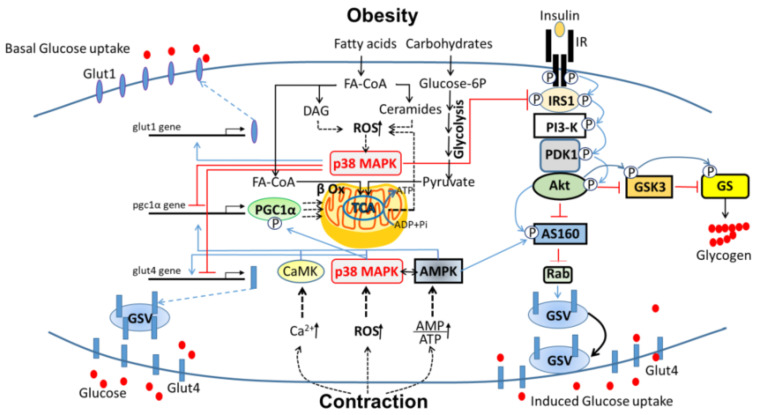
The involvement of p38 mitogen-activated protein kinase (MAPK) in glucose metabolism of skeletal muscle in health and disease. The upper half of the scheme describes the involvement of p38 MAPK in glucose metabolism of skeletal muscle in obesity: excessive intake of fatty acids and carbohydrates cause mitochondrial electron leak from the electron transport chain (ETC), accumulation of ROS, and mitochondrial dysfunction. Intramuscular fat metabolites (ceramide and DAG) reduce mitochondrial oxidative capacity and increase in the generation of mitochondrial reactive oxygen species (ROS) that induce p38 MAPK activity. Activated p38 MAPK inhibits IRS1 of insulin signaling through inhibitory phosphorylation. It also inhibits the transcription of *Pgc1α, Glut4* genes, and activates that of the *Glut1* gene. As a result, insulin-dependent glucose uptake is blocked, while insulin-independent glucose uptake is elevated. The lower half of the scheme describes the involvement of p38 MAPK in glucose metabolism of skeletal muscle in exercise: transient elevation in ROS induces the activity of p38 MAPK, which in turn stimulates the transcription of *Pgc1α* and *Glut4* genes. Besides, p38 phosphorylates PGC1α and augments its activity needed for mitochondrial integrity and function. P38 MAPK synergizes with AMPK in glucose uptake; the first increases the levels of Glut4 and the second drives the transport of vesicular Glut4 to the plasma membrane. Abbreviations: GSV, Glut4 storage vesicles; DAG, diacylglycerol; PI3-K, Phosphoinositide 3-kinase; PDK1, Phosphoinositide-dependent protein kinase-1; GSK3, Glycogen synthase kinase 3; GS, Glycogen synthase; IR, Insulin receptor; FA, Fatty acid.

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
