# Peer review of "p38 MAPK in Glucose Metabolism of Skeletal Muscle: Beneficial or Harmful?"

_ijms, 2020, doi:10.3390/ijms21186480_

Round 1

Reviewer 1 Report

This is generally a well-written and concise review detailing some of the evidence that supports the dual role of p38 MAPK in glucose metabolism in skeletal muscle. However, there are a number of important references which have been omitted, which I believe gives a more balanced view to the “good” and “bad” debate of p38 MAPK in glucose metabolism. I also think it’s important to provide more emphasis that p38 MAPK is also likely involved in mediating insulin-dependent (not just insulin-independent) glucose uptake. I believe that after including and discussing these important papers, this review will provide a better balanced perspective and hopefully help shift the focus of future studies to explore both “good” and “bad” aspects of SAPK/MAPK signalling in health and disease.

General Comments:

  1. Introduction
  • Second sentence of the Introduction, I am not sure “glucose consumption” is the most accurate term to use here. I would recommend to change to “glucose disposal” or “glucose uptake”.
  • Could also consider changing “is imported from the blood” to “is extracted from the blood”.

2.1 Glucose uptake:

  • In reference to this sentence “In addition, accumulation of cytosolic reactive oxygen species (ROS) produced by NADPH oxidase 2, entailed phosphorylation of p38 MAPK and stimulated Glut4-dependent glucose uptake in muscle during exercise [50]”. This group concluded that p38 MAPK phosphorylation was not consistent between their various conditions tested, and therefore was unlikely to explain their findings. This should be discussed as it is an important finding, and provides a more balanced perspective on the ambiguity behind p38 MAPK and glucose regulation.
  • Although it seems the focus is mainly on contraction-mediated glucose uptake – in the context of insulin-resistance (discussed in a later part) I think it is important to further discuss the potential role of p38 MAPK in enhancing insulin-mediated glucose uptake, with or without prior exercise. For example, Parker et al. found that insulin stimulation increases p38 MAPK phosphorylation in human skeletal muscle; Exercise also increases p38 MAPK phosphorylation; and exercise + insulin induces even greater 38 MAPK phosphorylation which was correlated with enhanced insulin sensitivity after exercise. Trewin et al. also measured p38 MAPK after exercise and insulin stimulation, with and without NAC supplementation. Research by Thong et al. also looked at exercise, insulin stimulation, and p38 MAPK. Additional work by Somwar et al could also be discussed (alongside their other findings that are already mentioned). These findings are important and should be mentioned/discussed in this section to provide greater balance to the “good” versus “bad” debate.

Ref: Parker, L., N.K. Stepto, C.S. Shaw, et al., Acute High-Intensity Interval Exercise-Induced Redox Signaling Is Associated with Enhanced Insulin Sensitivity in Obese Middle-Aged Men. Front Physiol, 2016. 7(411): p. 411.

Trewin, A.J., L.S. Lundell, B.D. Perry, et al., Effect of N-acetylcysteine infusion on exercise-induced modulation of insulin sensitivity and signaling pathways in human skeletal muscle. Am J Physiol Endocrinol Metab, 2015. 309(4): p. E388-97.

Somwar, R., D.Y. Kim, G. Sweeney, et al., GLUT4 translocation precedes the stimulation of glucose uptake by insulin in muscle cells: potential activation of GLUT4 via p38 mitogen-activated protein kinase. Biochem J, 2001. 359(Pt 3): p. 639-49.

Thong, F.S., W. Derave, B. Urso, et al., Prior exercise increases basal and insulin-induced p38 mitogen-activated protein kinase phosphorylation in human skeletal muscle. J Appl Physiol (1985), 2003. 94(6): p. 2337-41.

3.1 insulin resistance:

  • In reference to the sentence “Skeletal muscle accounts for about 80% of insulin-mediated glucose uptake in peripheral tissues and is the primary organ regulating glucose balance [1, 2].” – This is only during insulin clamp (i.e., hyperinsulinemic) conditions? Skeletal muscle glucose uptake during postprandial conditions is closer to 26% (Meyer et al. 2002). Please amend to make this more accurate and clearer for the reader.

Ref: Meyer, C., J.M. Dostou, S.L. Welle, et al., Role of human liver, kidney, and skeletal muscle in postprandial glucose homeostasis. Am J Physiol Endocrinol Metab, 2002. 282(2): p. E419-27.

  • Minor typo: “Intracellular lipid metabolites  increase oxidative stress and the generate reactive oxygen species (ROS) “that” inhibit insulin action via activation of serine/threonine kinases and serine phosphorylation of insulin receptor substrate 1 (IRS1) [68].”
  • It’s not just lipids that lead to ROS and insulin resistance, carbohydrates can also lead to electron leak in the mitochondria which leads to ROS production and insulin resistance (Fisher-Wellman et al 2012; Tucker et al. 2008).

Ref: Fisher-Wellman, K.H. and P.D. Neufer, Linking mitochondrial bioenergetics to insulin resistance via redox biology. Trends Endocrinol Metab, 2012. 23(3): p. 142-53.

Tucker, P.S., K. Fisher-Wellman, and R.J. Bloomer, Can exercise minimize postprandial oxidative stress in patients with type 2 diabetes? Curr Diabetes Rev, 2008. 4(4): p. 309-19.

  • In reference to the sentence “Interestingly, despite  its  role  in  preventing  insulin-mediated  glucose  uptake,  p38  MAPK  may increase  basal  and  insulin-independent  glucose  uptake  by  skeletal  “ – In light of current evidence I would also suggest that p38 MAPK may also play a role in insulin-dependent glucose uptake. I think this should be mentioned here, and incorporated throughout the review.

3.3.   P38 MAPK as a potential target for prevention of obesity-induced T2DM: 

  • In reference to the sentence “The prevailing model is that mitochondrial dysfunction impairs muscle ability to oxidize fatty acids, resulting in muscle lipid accumulation and, consequently, in insulin resistance [67, 99].” – Excess carbohydrates (in the absence of ATP utilisation) also causes mitochondrial electron leak, ROS and insulin resistance.
  1. Conclusions.

The discussion of hyperglycaemia (i.e., carbohydrates), ROS and potential activation of p38 MAPK needs to be more prevalent throughout (including the figure). As it stands the review is one-sided and it comes across as if lipids are the only important substrate for causing ROS, SAPK/MAPK activation, and insulin resistance.

Author Response

Dear Editor,

Two referees reviewed the manuscript, one accepted it as is, while the second called for a minor revision. Below, please find our detailed response to the reviewer’s comments:  

Reviewer 1: In general, the reviewer thought that the article was well written and concise. Yet, the reviewer claims that it should be more balanced from the point of view that p38 also likely be involved in mediating insulin-dependent and not only insulin-independent glucose transport. The reviewer provided several manuscripts that dealt with this aspect and suggested to include those in the manuscript.

General comments: 1. Introduction: The reviewer suggested replacing the term “glucose consumption” with “glucose disposal.” We changed the text presented in line 54 as requested. The phrase “is imported from the blood” was replaced with “ is extracted from the blood” as requested (line 60).

2.1 glucose uptake:  The reviewer related to a study by Henriquez Olguin et al 2019, about cytoplasmic ROS production by NOX2 and regulation of glucose uptake during exercise. We wrote in the original manuscript: “Besides, accumulation of cytosolic reactive oxygen species (ROS) produced by NADPH oxidase 2 (NOX2), entailed phosphorylation of p38 MAPK and stimulated Glut4-dependent glucose uptake in muscle during exercise [50].” The reviewer rightfully commented that p38 MAPK phosphorylation was not always consistent with glucose uptake at different conditions.  We, therefore, added a sentence in line 192: “Yet, the levels of p38 MAPK phosphorylation were not always consistent with the reduction in glucose uptake in muscles of mice devoid of NOX2 activity, indicative that p38 may not be the mediator of glucose uptake due to accumulation of cytoplasmic ROS.”

Another issue raised by the reviewer is “ the potential role of p38 in enhancing insulin-mediated glucose uptake with or without prior exercise”. The reviewer referred us to particular references. We added a discussion of the potential role of p38 in insulin-mediated glucose uptake in the last section of chapter  3.2 P38 MAPK in skeletal muscle insulin resistance, lines 338-352. In this paragraph, we discuss studies that implicate a role of p38 in insulin-mediated glucose uptake, and others that involve the activity of p38 in exercise (muscle contraction) and its role in the increase of insulin sensitivity of skeletal muscle.

3.1 Insulin resistance:  The reviewer rightfully related to the sentence: “skeletal muscle accounts for about 80% of insulin-mediated glucose uptake in peripheral tissues….” (line 250)  and quoted studies that estimated skeletal muscle glucose uptake during postprandial conditions was around 30%. We rephrased the sentence to make the message more precise: “Skeletal muscle accounts for  ~ 30% of the resting metabolic rate in human (Zurlo F 1990, Meyer C 2002) and up to  80% of glucose disposal under insulin-stimulated conditions (DeFronzo RA 1981) and it is thus the primary organ regulating glucose balance [1, 2].”

The reviewer indicated that “not just lipids lead to ROS and insulin resistance. Carbohydrates can also lead to electron leak in the mitochondria, which leads to ROS production and insulin resistance”. Again, the reviewer’s remark is justified, and we completely changed the related section and added more references. The new paragraph appears between lines 253-266: “One model explaining the development of insulin resistance in skeletal muscle argues that positive caloric balance causes the accumulation of lipids within muscle cells [67]. Intracellular lipid metabolites damage mitochondrial activity and consequently increase the production of reactive oxygen radicals. Another model predicts that calorie surplus imposes maximal mitochondrial respiration and leakage of electrons from the electron transport chain, forming oxygen radicals (Fisher-Wellman KH 201). Oxidative stress inhibits insulin action via activation of serine/threonine kinases and serine phosphorylation of insulin receptor substrate 1 (IRS1) [68]. Excessive oxygen radicals that cannot be neutralized by cellular defense mechanisms inflict cellular damage by oxidizing proteins, fats, and carbohydrates. The activity of p38 MAPK and other stress-activated kinases is induced by oxidative stress and is associated with insulin resistance (Henriksen EJ 2011).”

The reviewer suggested that we will add a sentence to the summary of “ P38 MAPK in skeletal muscle insulin resistance” that will mention the potential role of p38 MAPK in insulin-dependent glucose uptake. We added such a sentence in line 360: “ Acute activation of p38 MAPK may also play a role in insulin-dependent glucose uptake.”

3.3 P38 MAPK as a potential target for prevention of obesity-induced T2DM:   The sentence in line 403 was modified according to the reviewer’s suggestion from: “ The prevailing model is that mitochondrial dysfunction impairs muscle ability to oxidize fatty acids….”  To:  “The prevailing model is that excessive calorie intake cause mitochondrial electron leak, accumulation of ROS, mitochondrial dysfunction, and  insulin resistance [67, 99].” 

The reviewer requested that the role of carbohydrates, generation of ROS, which activates p38 MAPK, will be emphasized in the concluding part, including the figure. In response to the reviewer’s comment, we changed the text found within lines 439 to 442 to: “The surplus in skeletal muscle calorie intake and the elevated levels of circulating fatty acids that penetrate muscle cells, increase intramuscular fat metabolites and reduce mitochondrial oxidative capacity and increase in the generation of ROS.” We also added some changes to the figure that show that oxidative stress involved in insulin resistance is generated by mitochondrial dysfunction due to positive calorie balance, not necessarily due to fatty acids, but also due to excess of carbohydrates.

We believe that the reviewer remarks were fully considered in our revised manuscript and hope that it will be accepted for publication.  

Reviewer 2:  The reviewer accepted the manuscript as is and did not ask for any changes.

Both reviewers found that the English and style were fine. We edited the manuscript once again and made spelling and English corrections.

Sincerely yours,

Eyal Bengal, PhD

Department of Biochemistry

Technion-Israel Institute of technology

Reviewer 2 Report

This review provides a comprehensive overview of research reports on the involvement of p38 MAPK in skeletal muscle glucose metabolism in health conditions or disease, and is expected to provide comprehensive information to researchers in the field. The manuscript is well organized and is ready for publication as is.

Author Response

The reviewer accepted the manuscript as is and did not ask for any changes.